# Clustering-Based Multi-Region Coverage-Path Planning of Heterogeneous UAVs

Peng Xiao [1], Ni Li [1,2,*], Feng Xie [1,3], Haihong Ni [1], Min Zhang [1] and Ban Wang [1]

1 School of Aeronautics, Northwestern Polytechnical University, Xi'an 710072, China;
xiaop@mail.nwpu.edu.cn (P.X.); xiefeng_new@mail.nwpu.edu.cn (F.X.); nihaihong@mail.nwpu.edu.cn (H.N.);
2021200130@mail.nwpu.edu.cn (M.Z.); wangban@nwpu.edu.cn (B.W.)
2 Xi'an Aircraft Intelligent Recognition and Control Key Lab, Xi'an 710072, China
3 Aviation Industry Corporation of China, Chengdu Aircraft Design and Research Institute,
Chengdu 610091, China
* Correspondence: lini@nwpu.edu.cn

**Abstract:** Unmanned aerial vehicles (UAVs) multi-area coverage-path planning has a broad range of applications in agricultural mapping and military reconnaissance. Compared to homogeneous UAVs, heterogeneous UAVs have higher application value due to their superior flexibility and efficiency. Nevertheless, variations in performance parameters among heterogeneous UAVs can significantly amplify computational complexity, posing challenges to solving the multi-region coverage path-planning problem. Consequently, this study studies a clustering-based method to tackle the multi-region coverage path-planning problem of heterogeneous UAVs. First, the constraints necessary during the planning process are analyzed, and a planning formula based on an integer linear programming model is established. Subsequently, this problem is decomposed into regional allocation and visiting order optimization subproblems. This study proposes a novel clustering algorithm that utilizes centroid iteration and spatiotemporal similarity to allocate regions and adopts the nearest-to-end policy to optimize the visiting order. Additionally, a distance-based bilateral shortest-selection strategy is proposed to generate region-scanning trajectories, which serve as trajectory references for real flight. Simulation results in this study prove the effective performance of the proposed clustering algorithm and region-scanning strategy.

**Keywords:** heterogeneous UAVs; multi-region; coverage-path planning; regional allocation; clustering algorithm

## 1. Introduction

With the rapid development of artificial intelligence and automated control [1–3], UAVs have been widely used in both military and civilian fields, including reconnaissance and strike operations [4,5], target tracking [6–9], forest fire prevention [10], regional surveillance [11–13], etc. Due to limitations in performance and payload capacity, it is often challenging for one UAV to complete complex missions [14]. Therefore, the study of multi-UAV systems, which have good scalability and cooperation capabilities, has gained significant attention in current research. In order to maximize the overall effectiveness of the multi-UAV system, it is necessary to research task planning to obtain suitable task orders and flight paths of UAVs.

As an important branch of multi-UAV task planning, coverage-path planning includes region allocation and path planning. It has been studied by numerous scholars from various aspects, such as region shape [15,16], energy constraints [17–20], and obstacle avoidance [21,22]. Nielsen [15] tackled the issue of region coverage for non-convex polygons by dividing the area into numerous separate convex sub-polygons and utilizing a scanning pattern to ensure complete coverage. Huang [17] determined the energy consumption of UAVs in various flight modes and presented a coverage path-planning algorithm

that relies on a UAV energy-limited model, aiming to minimize the flight duration of UAVs on coverage paths. Franco [19] takes into account additional factors, including energy, speed, acceleration, and image resolution, in coverage-path planning. He proposes an energy model based on accurate measurements and utilizes it to develop a coverage path-planning algorithm that simultaneously achieves low power consumption and desired image resolution. Maza and Ollero [21] decomposed the entire region into multiple sub-regions and individually assigned them to the UAVs according to the flight and energy capabilities. They utilized the back-and-forth method to cover the sub-regions with the principle of minimizing the number of turning maneuvers.

Although the aforementioned studies have effectively addressed the issue of coverage-path planning, they primarily focus on the collaborative coverage of a single area using multiple UAVs and are concerned about how to divide the entire region. Therefore, these methods cannot be directly applied to the planning problem of multiple UAVs covering multiple areas. To address the multiple UAVs covering multiple areas problems, Mou et al. [23] utilized a deep reinforcement learning approach to match all areas with UAVs and developed a novel coverage path-planning algorithm based on the star communication topology to achieve comprehensive scanning of all areas. In [24], to study the penguins in various regions of Antarctica using UAVs, Shah presented a path-planning algorithm called Path Optimization for Population Counting with Overhead Robotic Networks, which exhibited faster computational speed in comparison to Mixed Integer Linear Programming (MILP) of the same size. Li [25] achieved the scan coverage of large-scale target areas by establishing an extended model of the Traveling Salesman Problem, with the optimization goals of coverage rate and completion time, while considering the performance constraints of UAVs. However, the above studies only focused on homogeneous UAVs and did not consider coverage-path planning for heterogeneous UAVs.

Compared to homogeneous UAV systems, the heterogeneous system exhibits greater flexibility and adaptability when it comes to complex tasks, thus improving the overall efficiency of the system [26,27]. However, the diverse range of UAV types makes designing a scheme that effectively leverages the capabilities of each UAV challenging. This challenge is especially prominent for multi-UAV collaborative decision-making, task planning, and formation control tasks. Moreover, in the problem of coverage-path planning, the heterogeneity among UAVs and the existence of multiple sub-regions also magnify the scale of the problem [28] and intensify the challenge of finding solutions [29,30]. Chen et al. [31] studied the multi-area coverage-path planning of multiple UAVs. The authors achieved clustering of sub-areas by calculating the density of sub-areas, but the visiting order of UAVs to the areas was not taken into consideration. In [32], Chen proposed an ant colony system (ACS)-based algorithm that achieves the area allocation using an effective time ratio and optimizes the UAV access sequence using the ant colony algorithm. However, this method faces a conflict between the optimization objectives of area allocation and sequence optimization, and it ignores the study of the actual flight trajectory of UAVs.

According to the publicized study, this type of research typically encounters the problem of repetitive sorting during the allocation and regional ordering optimization processes, which leads to a decrease in the accuracy of planning outcomes. They do not provide a specific method for planning the actual flight reference trajectory for UAVs. This study investigates the problem of multi-area coverage-path planning of heterogeneous UAVs with varying flight and scanning capabilities. This study aims to determine the optimal scan sequence and flight trajectories for UAVs to different areas, ensuring lower computational complexity and higher accuracy, considering constraints such as UAV maneuverability limitations and task requirements. The structure of this study is shown in Figure 1. This framework demonstrates how this study tackles the problem of multi-region coverage task planning of multi-UAVs. The primary contributions of this study are as follows:

1.　An optimization model is employed to tackle the problem of multi-region coverage-path planning of heterogeneous UAVs, and it is formulated using integer linear

programming. The problem of multi-area coverage-path planning is divided into two subproblems: allocating the task regions and determining the execution order. This decomposition significantly reduces the complexity associated with solving the problem;

2.  Based on the iterative idea of the k-means algorithm and the requirement of multi-region coverage, a novel clustering algorithm based on spatiotemporal similarity is proposed, and a clustering center iteration method is designed to complete regions clustering;

3.  A novel method is proposed to minimize the flight distance of a single UAV when scanning multiple regions. The method involves selecting entry points and scanning patterns based on the shortest flight distances.

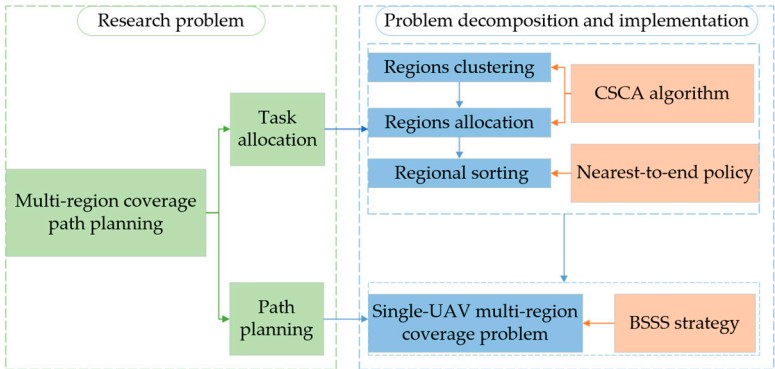

**Figure 1.** The structure of this study.

This study is organized as follows. Section 2 establishes the model and formulas for the multi-region coverage path-planning problem of heterogeneous UAVs. Section 3 proposes a novel region clustering algorithm and optimizes the visited order of regions. Section 4 introduces a trajectory-planning method based on the shortest flight distance. Section 5 presents simulation experiments and comparative analysis. Finally, Section 6 concludes this work.

## 2. System Model

The coverage path-planning problems of heterogeneous UAVs are often classified as non-deterministic polynomial hard (NP-hard); it is difficult to obtain an accurate solution directly [33]. So, we first analyze the constraints that need to be satisfied when the UAV system performs tasks and obtain the exact formulation. Then, in order to improve the solution efficiency, we intend to solve this problem from two aspects: regional allocation and regional sequence optimization, which are achieved through regional clustering and subregion reordering, respectively. Finally, we present a multi-region coverage-path planning strategy specifically for the practical flight trajectory of UAVs.

### 2.1. Problem Description

This study deploys $n$ fixed-wing heterogeneous UAVs $U = \{U_1, U_2, \ldots, U_n\}$ to carry out reconnaissance and scanning missions across $m$ rectangular regions $R = \{R_1, R_2, \ldots, R_m\}$; the task scenario diagram is shown in Figure 2. A list of key symbols used in this study and their definitions are provided in Table 1. In this work, the main differences between the UAVs are their flight speed, endurance, and scanning width. So, each UAV is characterized as $U_i = \langle Uid_i, Upos_i, V_i, T_i, d_i \rangle$, where $Uid_i$ and $Upos_i$ represent the serial number and the takeoff coordinate of $U_i$, respectively. $V_i$ denotes the cruising flight speed, and we assume that the cruising for each UAV maintains a constant throughout the mission. $T_i$ represents the maximum endurance for cruising flight, while $d_i$ represents the scan width of sensors installed in $U_i$.

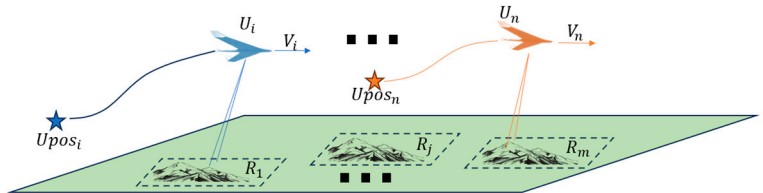

**Figure 2.** Task scenario diagram.

**Table 1.** Symbol definitions.

| Symbol | Definitions |
|---|---|
| $U$ | The set of UAVs |
| $U_i$ | The $i$th UAV in the set $U$ |
| $n$ | The number of UAVs in the set $U$ |
| $Upos_i$ | The takeoff position of $U_i$ (also called the base of $U_i$) |
| $V_i$ | The cruising flight speed of $U_i$ |
| $T_i$ | The maximum flight endurance of $U_i$ |
| $d_i$ | The scan width of onboard sensors of $U_i$ |
| $R$ | The set of regions |
| $R_j$ | The $j$th region in the set $R$ |
| $m$ | The number of regions in the set $R$ |
| $Rpos_j$ | The center coordinate of $R_j$ |
| $Rbeta_j$ | The angle between the major axis and the X-axis of the map |
| $L_j$ | The length of the major axis of $R_j$ |
| $W_j$ | The length of the minor axis of $R_j$ |
| $D_{j,k}$ | The distance between $R_j$ and $R_k$ |
| $TF_{i,j,k}$ | The time consumption of $U_i$ in flying from $R_j$ to $R_k$ |
| $TS_{i,j}$ | The time consumption of $U_i$ in scanning $R_j$ |
| $C$ | The set of clusters |
| $C_i$ | The $i$th cluster in the set $C$ |
| $STS_{i,j}$ | The spatiotemporal similarity between $C_i$ and $R_j$ |
| $Que_i$ | The regions queue of $C_i$ |
| $Seq_i$ | The sequence of regions that $U_i$ is going to coverage |
| $\{Lin, Win\}$ | The optional entry points set of a region |

In this study, each region's features are shown as $R_j = \langle Rpos_j, Rbeta_j, L_j, W_j \rangle$. In this notation, $Rpos_j$ represents the coordinates of the center point in the rectangular region. $Rbeta_j$ signifies the angle formed between the major axis of the region and the X-axis of the map, with a counterclockwise direction identified as positive. $L_j$ represents the length of the major axis, and $W_j$ denotes the length of the minor axis.

An $m$-row $m$-column matrix $D = \left\{ D_{j,k} \right\}$ is adopted to represent the distance between regions $R_j$ and $R_k$, which is calculated using the Euclidean distance between their center points. In this study, heterogeneous UAVs are required to travel from their takeoff positions $Upos_i$ to the designated mission area. Upon finishing their assigned area scanning tasks, the UAVs must return to their respective starting points. As a result, $TF_{i,j,k}$ is used to represent the time consumption of $U_i$ flying from one region $R_j$ to another region $R_k$. Since we do not know the order in which the UAV scans regions during the allocation phase, the flying distance is roughly estimated using the distance between the centers of these two regions. $TS_{i,j}$ is used to represent the time consumption of $U_i$ scanning the region $R_j$; each UAV will use the back-and-forth scanning method to scan a single area. $TF_{i,j,k}$ and $TS_{i,j}$ can be calculated using

$$TF_{i,j,k} = \frac{D_{j,k}}{V_i} \tag{1}$$

$$TS_{i,j} = \frac{L_j \times W_j}{V_i \times d_i} \tag{2}$$

*2.2. Exact Formulation*

This section analyzes the constraints of the multi-area coverage task-allocation problem of heterogeneous UAVs, and mathematical formulations for each constraint are established. Furthermore, an optimization model for task allocation is established, aiming to minimize the overall completion time of the system.

To obtain appropriate allocation results, the main constraints in this study are as follows:

C1: The constraint on the takeoff and landing of UAV. Once a UAV is selected to perform the scanning task, it needs to depart from the starting point and return to the starting point after completing all tasks.

C2: The constraint on the scanning of regions. To avoid performing things twice, each task region should only be scanned by one UAV. It also means that only one UAV can fly to and from one mission area. This will prevent repeated scanning of the same area.

C3: The constraint on the maximum endurance of an UAV. During the execution of missions, the total flight time $U_i$ cannot exceed its maximum endurance requirement.

C4: The constraint on the number of regions covered by a single UAV. The number of scanning regions conducted by each UAV must not surpass the total number of regions.

C5: The constraint on the number of UAVs engaged in all tasks. This constraint ensures that the number of UAVs carrying out the task remains below the total quantity and is greater than one.

Constraint (C1) refers to the restriction that all UAVs can only perform takeoff and landing operations at most once. This also means that if the UAV departs from its base, it is obliged to return to the base upon completion of the coverage task. To describe constraint (C1) through mathematical expressions, this study adopts a Boolean array $X = \left\{ x_{i,j,k}, 1 \leq i \leq n, 0 \leq j \leq m, 0 \leq k \leq m \right\}$ to represent the decision variables of UAVs' planned paths, where the subscripts $j, k$ represent $U_i$ flies from $R_j$ to $R_k$. It is worth noting that only if $U_i$ is chosen to fly from $R_j$ to $R_k$, Boolean variable $x_{i,j,k} = 1$, otherwise, $x_{i,j,k} = 0$. If $j = 0$ or $k = 0$, it represents that the $U_i$ departs from or returns to the starting point. Through the aforementioned description, constraint (C1) can be expressed as:

$$\forall i \in [1,n], \begin{cases} \sum_{j=1}^{m} x_{i,0,j} = \sum_{j=1}^{m} x_{i,j,0} = 1, U_i \textit{ is chosen to execute task} \\ \sum_{j=1}^{m} x_{i,0,j} = \sum_{j=1}^{m} x_{i,j,0} = 0, \textit{otherwise} \end{cases} \tag{3}$$

Constraint (C2) pertains to the situation in which only one UAV is capable of traversing a given region. Therefore, constraint (C2) can be represented as:

$$\forall j \in [1,m], \sum_{i=1}^{n} \sum_{k=0}^{m} x_{i,j,k} = \sum_{i=1}^{n} \sum_{k=0}^{m} x_{i,k,j} = 1 \tag{4}$$

Constraint (C3) indicates that the overall flight duration of a chosen UAV, including departure from and return to the base and coverage of the specified task regions, should not exceed its maximum flight time, i.e.,

$$\forall i \in [1,n], \sum_{j=0}^{m} \sum_{k=0}^{m} x_{i,j,k} \times TF_{i,j,k} + \sum_{j=1}^{m} y_{i,j} \times TS_{i,j} \leq T_i \tag{5}$$

The variable $y_{i,j}$ in the above expression is a Boolean variable, which has a similar meaning with $x_{i,j,k}$. If $U_i$ needs to perform a coverage scanning task in $R_j$, $y_{i,j} = 1$, otherwise $y_{i,j} = 0$. The relationship between $y_{i,j}$ and $x_{i,j,k}$ can be expressed as:

$$\forall k \in \leq [0,m], x_{i,j,k} = y_{i,j} \tag{6}$$

Constraint (C4) implies that each UAV cannot perform more coverage tasks than the total number of regions, i.e.,

$$\forall i \in [1, n], \sum_{j=1}^{m} y_{i,j} \leq m \tag{7}$$

Constraint (C5) shows that the number of UAVs taking off from the bases to carry out the tasks must not exceed the total number of UAVs, which can be written as:

$$\sum_{i=1}^{n} \sum_{j=1}^{m} x_{i,0,j} \leq n \tag{8}$$

This article aims to seek the sequence of UAV access to the task regions in the scenario where heterogeneous UAV systems take off from different bases, scan and cover corresponding areas, and return to the bases. The goal is to minimize the overall system's task completion time $f$, and the constraints are constraints from (C1) to (C5). Since each UAV takes off simultaneously and flies towards their respective task regions, the task completion time of the entire system can be equivalently represented as the time consumed by the UAV, which returns to the base the last. Therefore, the optimization objective function $f$ can be expressed as:

$$f = \max_{1 \leq i \leq n} \left( \sum_{j=0}^{m} \sum_{k=0}^{m} x_{i,j,k} \times TF_{i,j,k} + \sum_{y=1}^{m} y_{i,j} \times TS_{i,j} \right) \tag{9}$$

The exact formulation of this system can be written as:

$$\begin{aligned} \min \quad & f \\ s.t. \quad & C1, C2, C3, C4, C5 \end{aligned} \tag{10}$$

In the above programming, the unknown variables are a mixture of integer (e.g., elements of $X$) and real variables (e.g., the maximum time $f$), and all constraints are linear. Therefore, this problem belongs to the class of MILP problems. Since the coverage-path planning of heterogeneous UAVs is NP-hard, although the precise solutions can be obtained using the proposed MILP formulation, it requires searching the entire solution space. Furthermore, the computational time will drastically increase with the growth of both the number of UAVs and the number of regions, resulting in significant consumption of computational time and cost. Inspired by the concept of clustering, we devised a clustering-based approach to tackle the task-allocation problem during the coverage-path planning of heterogeneous UAVs in the following sections. Implementing region allocation through clustering, and then optimizing the region access sequence, can greatly reduce the complexity of the problem.

### 3. Coverage Scanning Clustering Algorithm-Based Coverage-Path Planning

The target clustering methods normally consist of two steps: target area clustering and cluster target allocation. Target area clustering involves clustering the regions based on specific characteristics; cluster target allocation requires assigning each sub-cluster to a specific UAV.

The k-means clustering algorithm is one of the iterative and classical target clustering methods. It has the advantages of simplicity, wide applicability, and fast convergence speed. Therefore, it is frequently applied to solve multi-UAV task assignment problems [34–36]. The basic process of the k-means clustering algorithm is commonly represented as:

Step 1: Determine a value $k$, which represents the number of sub-clusters aiming to obtain by clustering the dataset.

Step 2: Randomly select $k$ data points from the dataset as centroids (the centers of the sub-clusters).

Step 3: For each point in the dataset, calculate its distance to each centroid and assign it to the sub-cluster with the nearest centroid.

Step 4: Calculate the mean coordinates of all points within the sub-cluster and take it as the new centroid.

Step 5: If the distance between the newly calculated centroid and the previous centroid is less than a predefined threshold, it can consider the clustering to have achieved the desired result, and the algorithm terminates. Otherwise, we need to iterate Step 3~Step 5.

From the flow of the k-means clustering algorithm, it can be seen that it is highly sensitive to the initial selection of centroids, and different random seeds can yield completely different clustering results, significantly influencing the outcome. Moreover, the k-means clustering algorithm focuses on clustering data points, while the research background of this study involves multiple heterogeneous UAVs flying to multiple rectangular regions to perform scanning tasks. Therefore, the dataset in this study consists of planes rather than points, making the k-means algorithm unsuitable for this study. Inspired by the iterative clustering idea of the k-means algorithm, this study proposes an algorithm called the coverage-based scanning clustering algorithm (CSCA), which includes three major phases: the initial cluster centers selection phase, the multi-regional initial clustering phase, and the clustering regions update phase. Additionally, we investigated the method for optimizing the access order of regions following the completion of the clustering process.

### 3.1. Initial Cluster Centers Selection Phase

The purpose of clustering the various regions is to allocate each clustered subset area to its corresponding UAVs to complete the overall task in the shortest possible time. This means that each UAV will be assigned to a sub-clustered area. Afterward, the UAVs will perform scanning tasks of the regions within their respective sub-clusters in a certain order.

The task completion time includes both flight time and scanning time. The clustering centers have spatiotemporal similarities (which will be explained in the second phase) with their corresponding subsets of regions. Consequently, the duration of the UAV flight time is partially influenced by the distance between the UAV base and the clustering center. In other words, the shorter the distance between the clustering center and the UAV base, the less time the UAV will spend on flight. Therefore, we take the bases of each UAV as the initial clustering center points for the CSCA algorithm. A set $C = \{C_1, C_2, \ldots, C_n\}$ is used to store the clustering centers, where the serial number of cluster centers is equal to the serial number of UAVs involved in the given tasks. Each clustering center is defined as $C_i = \langle x_i, y_i \rangle$, where $x_i$ and $y_i$ represent the coordinates on the X-axis and Y-axis of the map, respectively. For each $C_i$, a queue $Que_i = \{que_i^1, que_i^2, \ldots, que_i^e\}$ is utilized to indicate the indexes of regions that have been grouped into $C_i$ and would be scanned by UAV $U_i$; additionally, $e$ represents the total number of regions to be covered by $U_i$.

### 3.2. Multi-Regional Initial Clustering Phase

After determining the initial cluster centers, the initial clustering methods for the regions will be proposed in this phase. A $n$-row and $m$-column matrix $STS = \{STS_{i,j}\}$ is employed to characterize the connection between the regions targeted for clustering and the clustering centers. $STS_{i,j}$ is defined as a spatiotemporal similarity between cluster $C_i$ and region $R_j$. It consists of two parts: the time taken by $U_i$ to travel from the center $R_j$ to the cluster center $C_i$, and the time taken by $U_i$ to coverage $R_j$, i.e.,

$$STS_{i,j} = TS_{i,j} + \frac{d(R_j, C_i)}{V_i} \tag{11}$$

The above equation $d(R_j, C_i)$ refers to the Euclidean distance between the regional center $R_j$ and the cluster center $C_i$. It is necessary to calculate the spatiotemporal similarity of each $R_j$ with all $C_i$ in the set $C$ to find the minimum value index $ind_i$, and add the index $j$ corresponding to $R_j$ to the queue $Que_{ind_i}$, i.e.,

$$\forall j \in [1, m], \begin{cases} ind_i = \mathrm{argmin}(STS_{i,j}) \\ Que_{ind_i} = Que_{ind_i} \cup j \end{cases} \tag{12}$$

### 3.3. Clustering Regions Update Phase

This phase is the core of the proposed CSCA algorithm, where the cluster centers will continuously change with each iteration until the desired outcome is achieved. For the cluster $C_i$, the mean coordinates of all regional centers in this cluster are defined as the updated coordinates for the cluster center. Additionally, $TR(C_i)$ is used as an approximate substitute for the task-completion time requirements of $C_i$, which can be written as,

$$TR(C_i) = 2 \times d(Upos_i, C_i) + \sum_{l=1}^{e} STS_{i,que_i^l} \tag{13}$$

From the above formulation, it can be deduced that the more regions $C_i$ contains, the larger $TR(C_i)$ will be. Consequently, this will lead to an overall increase in the total task completion time for UAV $U_i$. Our goal in this study is to minimize the overall completion time of the entire heterogeneous UAV system, meaning that there is minimal difference in completion time between the UAVs. Thus, this can be equivalently represented as minimizing the difference between the maximum completion time $\max\{TR(C_i)\}$ and the minimum completion time $\min\{TR(C_i)\}$ of tasks.

In order to meet the maximum endurance constraint of UAVs, it is imperative to determine the remaining flight time $RFT(U_i)$ for each individual UAV $U_i$. $RFT(U_i)$ can be calculated via,

$$RFT(U_i) = T_i - TR(C_i) \tag{14}$$

If there is any $RFT(U_i)$ value less than 0, the $TR(C_i)$ of the $RTF(U_i)$ with the lowest negative value should be assigned as $\max\{TR(C_i)\}$. In order to reduce the regions of the cluster corresponding to $\max\{TR(C_i)\}$ and increase the regions of the cluster corresponding to $\min\{TR(C_i)\}$, we proposed a region-transfer strategy based on the ranking of clustering center distance, which consists of mainly three steps. In the following steps, to enhance convenience and facilitate ease of understanding, the symbol *maxind* and *minind* are employed as representations of $\arg(\max\{TR(C_i)\})$ and $\arg(\min\{TR(C_i)\})$, respectively:

Step 1: Calculate the distances between the cluster center $C_{maxind}$ and all the remaining cluster centers and sort them in ascending order to obtain the sequence $\Pi = \{\pi_1, \pi_2, \ldots, \pi_{n-1}\}$. $\Pi(i)$ is equal to $\pi_i$, and each $\pi_i$ represents the index number of a cluster center.

Step 2: Compute the distance between all regions $\{que_{maxind}^1, que_{maxind}^2, \ldots, que_{maxind}^e\}$ in the $Que_{maxind}$ and the cluster center $C_{\Pi(1)}$, then assign the region index with the minimum distance to the $Que_{\Pi(1)}$ of cluster $C_{\Pi(1)}$.

Step 3: If $\Pi(1) = minind$, it indicates the completion of the region transfer process, otherwise, define *maxind* as $\Pi(1)$, and remove $\Pi(1)$ from the sequence $\Pi$. Update $\Pi$ by the distance between $C_{maxind}$ and other remaining cluster centers, then proceed to Step 2.

The schematic diagram of the algorithmic process described above is illustrated in Figure 3. Once the clustered regions have been updated, recalculate the centroids for each cluster and $TR(C_i)$. The centroid coordinate of $C_i$ can be calculated by

$$C_i = \frac{\sum_{l=1}^{e} Rpos_{que_i^l}}{e} \tag{15}$$

Subsequently, compute the disparity between the $\max\{TR(C_i)\}$ and $\min\{TR(C_i)\}$ values and ascertain whether it falls below the designated threshold. If the disparity is less than the threshold, the clustering assignment for all regions is considered complete. If not, proceed with the above region-transfer strategy.

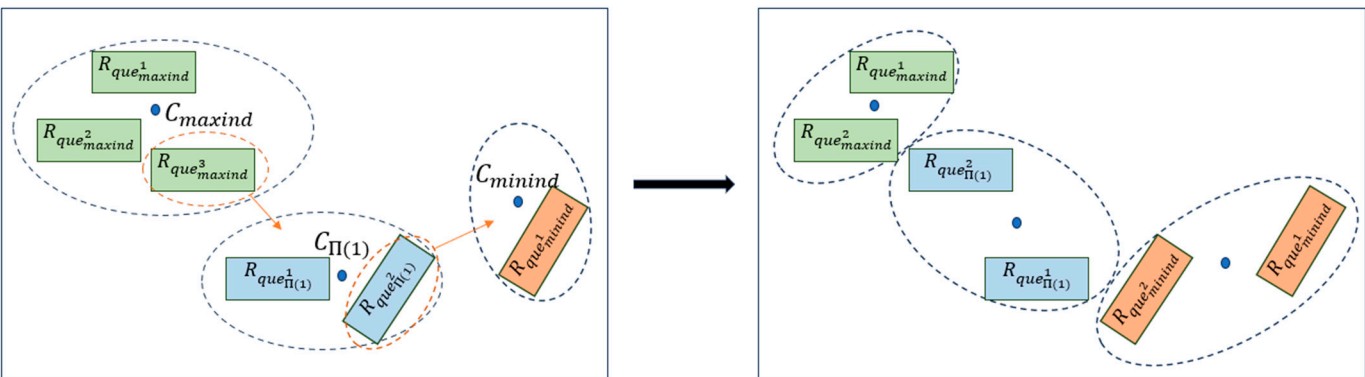

**Figure 3.** Illustrative diagram of region transfer.

To better understand the overall description of the coverage-based scanning clustering algorithm (indicated in Sections 3.1–3.3), the entire pseudo-codes of this algorithm are shown in Algorithm 1.

---

**Algorithm 1:** Pseudo-Codes of Coverage-Based Scanning Clustering Algorithm

---

**Input:** set of regions $R$, set of UAVs $U$
**Output:** the final set of clusters $C$ and regional queue $Que$
1: Take the base of each $U_i$ as the initial cluster center $C_i$;
2: Build $Que_i$ for each $C_i$ to store the index of regions;
3: **for** $i \leftarrow 1$ to $n$ **do**
4:　　**for** $j \leftarrow 1$ to $m$ **do**
5:　　　　Calculate $STS_{i,j}$ by Equation (11);
6:　　**end for**
7: **end for**
8: **for** $j \leftarrow 1$ to $m$ **do**
9:　　$ind_i \leftarrow \mathrm{argmin}\left(STS_{i,j}\right)$;
10:　　$Que_{ind_i} \leftarrow Que_{ind_i} \cup j$;
11: **end for**
12: **while** the difference between $\max\{TR(C_i)\}$ and $\min\{TR(C_i)\}$ has not dropped below the specified threshold **or** the maximum number of iterations has not been reached, **do**
13:　　Calculate the $TR(C_i)$ for each $C_i$ according to Equation (13) and determine $\max\{TR(C_i)\}$ and $\min\{TR(C_i)\}$;
14:　　Update the $RFT(U_i)$ for each $U_i$ according to Equation (14) and determine the minimum value $\min\{RFT(U_i)\}$;
15:　　**if** $\min\{RFT(U_i)\} < 0$ **do**
16:　　　Define the $TR(C_i)$ associated with $\min\{RFT(U_i)\} < 0$ as the $\max\{TR(C_i)\}$;
17:　　**end if**
18:　　Obtain the sequence $\Pi = \{\pi_1, \pi_2, \ldots, \pi_{n-1}\}$ according to the Step 1 of region-transfer strategy;
19:　　**while** $Que_{minind}$ and $Que_{maxind}$ have not completed the update, **do**
20:　　　Calculate the distance between all regions in the $Que_{maxind}$ and $C_{\Pi(1)}$;
21:　　　Assign the regional index $que^l_{maxind}$ with the minimum distance to the $Que_{\Pi(1)}$;
22:　　　**if** $\Pi(1) = minind$ **do**
23:　　　　**break**;
24:　　　**else do**
25:　　　　$maxind \leftarrow \Pi(1)$;
26:　　　　Remove $\Pi(1)$;
27:　　　**end if**
28:　　**end while**
29:　　Recalculate the centroid coordinate of each $C_i$ according to Equation (15)
30: **end while**

---

*3.4. Regional Sorting*

After executing the clustering assignment for all regions, if the number of regions in a sub-cluster is greater than one, it is necessary to sort the order of scanning regions. Therefore, the scanning order of regions assigned to UAVs will be addressed in this phase. There are many strategies that can be employed to solve this problem, such as the Genetic algorithm (GA), particle swarm optimization (PSO), and various other heuristic algorithms. However, these algorithms have the problems of non-uniqueness in calculation results and slow computation. Therefore, we adopt the nearest-to-end (NE) policy, which has significant efficiency and reliability. In this policy, all the unallocated regions in each $Que_i$ will be sorted. The policy involves two parts: initialization and interpolation.

(1)    Initialization

A sequence $Seq_i$ is used to denote the sequential order of tasks executed by $U_i$, and the region in the unallocated regions that is closest to the base $Upos_i$ is chosen as the tail ($seq_i^e$) of the sequence $Seq_i$. In addition, $Upos_i$ is utilized as the head ($seq_i^1$) of the sequence. Then, choose the unallocated region nearest to either the head or tail of the sequence, $R_{best}$, which can be represented as,

$$R_{best} = \text{argmin}\left(\min\left\{d\left(R_j, R_{seq_i^1}\right), d\left(R_j, R_{seq_i^e}\right)\right\}\right), R_j \in Que_{i,avail} \qquad (16)$$

(2)    Interpolation

After initialization, $R_{best}$ will be stored in $Seq_i$. If $R_{best}$ is closer to the tail than to the head, $R_{best}$ is placed in the end of $Seq_i$ and $Upos_i$ will be deleted from $Seq_i$; thus, $R_{best}$ will be the new tail; otherwise, put the $R_{best}$ in the first place of $Seq_i$ and delete $Upos_i$, thus $R_{best}$ becomes the new head. As a result, the unallocated regions in $Que_i$ can be sorted using Equation (16) and placed in the $Seq_i$ by the head-tail update method described above until all regions in $Que_i$ are allocated.

Once all the regions have been sorted, the mission duration of all UAVs can be calculated effortlessly using Equations (1) and (2), and the maximum duration signifies the overall mission duration of the entire heterogeneous UAV system. The pseudo-codes of the coverage-based scanning clustering algorithm with a nearest-to-end policy are shown in Algorithm 2, and the algorithm flowchart is shown in Figure 4.

---

**Algorithm 2:** Pseudo-codes of coverage-based scanning clustering algorithm with Nearest-to-End Policy

---

    **Input:** set of regions $R$, set of UAVs $U$
    **Output:** The regions scan sequence $Seq_i$ for each $U_i$
1: get the set of clusters $C$ and regional queue $Que$ according to Algorithm 1;
2: **for** $i \leftarrow 1$ to $n$ **do**
3:    **if** $e > 1$ **do**
4:       Initialize the $Seq_i$;
5:       the head of $Seq_i \leftarrow Upos_i$;
6:       the tail of $Seq_i \leftarrow$ the region in the unallocated regions that is closest to the base $Upos_i$;
7:       Obtain $R_{best}$ by Equation (16);
8:       Place $R_{best}$ in $Seq_i$ by the nearest-to-end policy and remove $Upos_i$;
9:       **while** there is any unassigned region in $Que_i$ **do**
10:         Calculate $R_{best}$ in unassigned regions;
11:         Insert $R_{best}$ in $Seq_i$;
12:       **end while**
13:    **else do**
14:       $Seq_i \leftarrow Que_i$;
15:    **end if**
16:    Calculate the time consumption of $U_i$;
17: **end for**

---

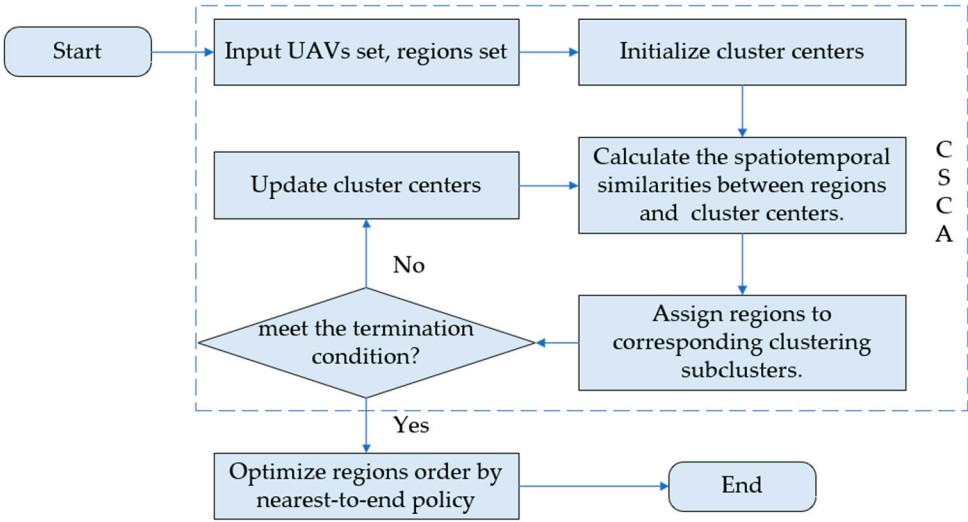

**Figure 4.** Algorithm flowchart.

## 4. Bilateral Shortest-Selection Strategy-Based Trajectory Planning

In the previous section, we discussed the multi-region allocation method and obtained the regional scanning sequence for each UAV. Hence, this section will introduce a path-planning method for the UAV flights.

The back-and-forth scanning method [21] is widely employed for regional scanning. When scanning a rectangular region $R_j$ using the back-and-forth scanning method, the scanning can be performed along either its long side or its short side. Currently, the majority of studies concentrate on scanning a single region. Consequently, scanning along the longer side indeed yields a shorter path compared to scanning along the shorter side. However, in this study, a single UAV must sequentially scan multiple regions. Its route encompasses the scanning distance within each region, as well as the flight distance between regions. As a result, scanning along the longer side does not necessarily minimize the overall distance, as it may increase the inter-region distance. Therefore, this study proposes a multi-region scanning path-planning method based on minimizing the total flight distance called the bilateral shortest-selection strategy (BSSS).

As defined earlier in this study, the scanning width of the sensors carried by $U_i$ is represented by $d_i$. Additionally, $R_j$ has a total of 8 optional entry points $\{Lin, Win\}$. They are classified into categories $\{Lin_1, \ldots, Lin_4\}$ and $\{Win_1, \ldots, Win_4\}$, depending on whether they are located on the long side or the short side, as shown in Figure 5. The red dots represent entry points on the long side, which means the UAV will scan along the shorter side of the region, while the blue dots represent entry points on the short side, which means the UAV will scan along the longer side of the region.

The current position of the UAV $U_i$ is defined as $Outpos_i$, and the areas to be scanned are $R_j$. The entry points $InL$ and $InW$ for the longer and shorter sides of region $R_j$ will be determined by calculating the closest points in $\{Lin_1, \ldots, Lin_4\}$ and $\{Win_1, \ldots, Win_4\}$ to $Outpos_i$, respectively, i.e.,

$$InL = \mathrm{argmin}(d(Outpos_i, Lin_1), \ldots, d(Outpos_i, Lin_4)) \tag{17}$$

$$InW = \mathrm{argmin}(d(Outpos_i, Win_1), \ldots, d(Outpos_i, Win_4)) \tag{18}$$

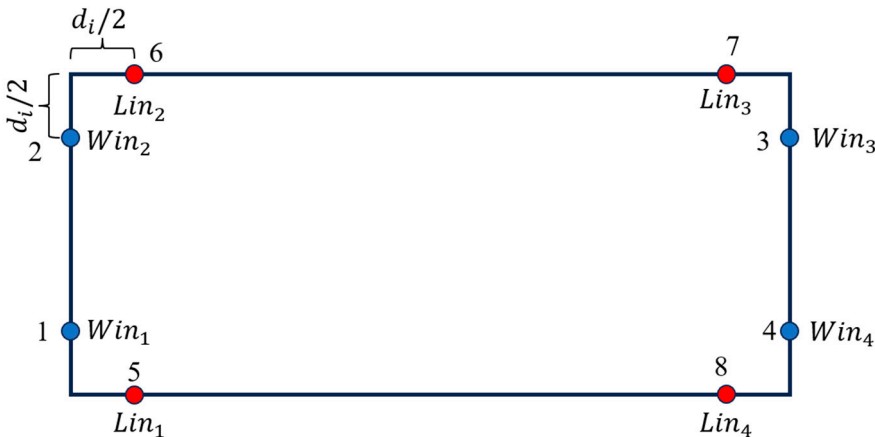

**Figure 5.** Illustrative diagram of scanning entry points in a rectangular region.

After defining the entry points of $R_j$, $U_i$ would fly from $Outpos_i$ to $InL$ or $InW$, using the back-and-forth scanning method and scanning along either the short or long sides. Upon completion of the scanning process, the departure points $OutL$ and $OutW$ will be identified for each region. Then, it is necessary to separately calculate the shortest distances between $OutL$ and $OutW$ with the optional entry points $\{Lin', Win'\}$ of the next to-be-scanned region. Therefore, the flying distance $Len\_W$ and $Len\_L$ of the aforementioned process can be represented individually as:

$$Len\_W = d(InL, Outpos_i) + \left\lceil \frac{L}{d} \right\rceil \times W + \left\lceil \frac{L}{d} - 1 \right\rceil \times \frac{\pi d}{2} + \frac{\min(d(OutL, Lin')) + \min(d(OutL, Win'))}{2} \quad (19)$$

$$Len\_L = d(InW, Outpos_i) + \left\lceil \frac{W}{d} \right\rceil \times L + \left\lceil \frac{W}{d} - 1 \right\rceil \times \frac{\pi d}{2} + \frac{\min(d(OutW, Lin')) + \min(d(OutW, Win'))}{2} \quad (20)$$

Since the subsequent region can be scanned either along its longer or shorter edge, the last terms in Equations (19) and (20) are expressed as a weighted average. If $Len\_L < Len\_W$, choose $InW$ as the entry point and scan along the longer side of the region. Otherwise, choose $InL$ as the entry point and scan along the shorter side of the region. Finally, the current exit point of the region will be defined as new $Outpos_i$, and we can employ the previously mentioned approach to determine both the entry point and the scanning method for the subsequent region.

## 5. Simulation Experiments

This section conducted numerical simulation experiments on the proposed CSCA algorithm and generated actual flight reference paths for UAVs based on the BSSS method. The simulation scenario was established, and subsequently, we validated the accuracy and effectiveness of our algorithm by conducting a comparative analysis with the spatial-temporal clustering-based algorithm (STCA) cited in [31], which provided an effective method for the assignment of heterogeneous UAVs multi-area scanning tasks. Utilizing the regional allocation outcomes obtained via CSCA, the actual feasible flight routes can be obtained using the BSSS method while accounting for the constraints imposed by UAV maneuverability.

### 5.1. Parameter Setting

For this simulation, the mission area is defined as a square with a side length of 50 km. There are a total of eight UAVs available for the mission, with their specific parameter settings presented in Table 2.

**Table 2.** Settings for UAV parameters.

| No. | 1 | 2 | 3 | 4 | 5 | 6 | 7 | 8 |
|---|---|---|---|---|---|---|---|---|
| Base coordinate (km, km) | (0, 10) | (0, 30) | (25, 0) | (10, 0) | (45, 0) | (25, 50) | (15, 50) | (50, 0) |
| Flying speed (m/s) | 35 | 45 | 35 | 50 | 45 | 40 | 45 | 45 |
| Maximum endurance (h) | 3.2 | 2.5 | 2.8 | 2.8 | 2.5 | 4.5 | 3.7 | 3.0 |
| Scanning width (m) | 650 | 600 | 750 | 650 | 650 | 650 | 600 | 500 |

The setting of the rectangular regions needs to obey the following rules:

1. Randomly assign coordinates to the center point of each rectangular region in order to achieve a uniform distribution across the entire map;
2. Set the range of values for the major axis $L \in [3, 3.5, 4]$ and the range of values for the minor axis $W \in [2, 2.5, 3]$, with units in kilometers;
3. Iterate through the rectangular region in order and assign values to the major and minor axes according to the index of the region. When the index is divisible evenly by 3, assign the values $L(1)$ and $W(1)$; when the index leaves a remainder of 1, assign the values $L(2)$ and $W(2)$; otherwise, assign the values $L(3)$ and $L(3)$;
4. Set the rotation angle of the rectangle. Divide $\pi$ into $m$ equal parts (i.e., $\pi/m$), where $m$ is the total number of regions. The rotation angle for each region is obtained by multiplying the index of the region by the equal parts angle (i.e., index $\times \pi/m$). The unit is radians, and it is defined that the counterclockwise direction is positive;
5. The total number of regions ranges from 5 to 40, with an increment of 5.

*5.2. Task Completion Time Simulation*

Under the premise of the same parameters and regional distribution of UAVs, we conducted a simulation comparison between the CSCA algorithm and the STCA algorithm in this section. The maximum task completion time of UAVs is used to represent the overall task completion time of the heterogeneous UAV system. Furthermore, we explored the reliability of the algorithm from the perspectives of different numbers of regions and different numbers of UAVs.

In Figure 6, we used two different region clustering and assignment algorithms, CSCA and STCA [31]. Region sequencing methods based on the nearest-to-end policy and genetic algorithm are used to compare the applicability of the CSCA and STCA for different region sequencing methods. The number of UAVs was three, and the X-axis represents the number of regions, with a step size of 5, while the Y-axis represents the task completion time of the UAV system. "STCA-NE", "CSCA-NE", "STCA-GA", and "CSCA-GA" correspondingly indicate the allocation of region-scanning sequences based on the STCA algorithm and the CSCA algorithm utilizing the nearest-to-end policy and allocation of region-scanning sequences based on the STCA algorithm and the CSCA algorithm employing the genetic algorithm. Under different region numbers, the statistical results of task completion time obtained using different algorithms are shown in Table 3.

From Figure 6, it can be observed that with the increase in the number of regions, the overall task completion time of the four methods shows an upward trend. Additionally, the CSCA-NE algorithm proposed in this study surpasses the STCA-NE algorithm. Furthermore, CSCA-GA consistently achieves the shortest task completion time, which demonstrates the outstanding performance of our clustering algorithm. Furthermore, it can be found that CSCA-GA and STCA-GA are superior to CSCA-NE and STCA-NE, respectively, due to the higher accuracy of genetic algorithms compared to the nearest-end strategy. From Table 3, it can further be observed that under different numbers of regions, when using the average task completion time as a measurement standard, CSCA-NE's result is 19.6% lower than STAC-NE, and CSCA-GA's result is 9.7% lower than STAC-GA. Moreover, the average difference between CSCA-NE and CSCA-GA is 60.6% lower than the average difference between STCA-NE and STCA-GA. This indicates that under different region quantities, the CSCA algorithm has a higher adaptability for the sequential allocation

of regions in clustering. As a result, the above analysis demonstrates that under the premise of a fixed number of UAVs scanning multiple regions, the CSCA algorithm exhibits higher reliability and accuracy compared to the STCA algorithm.

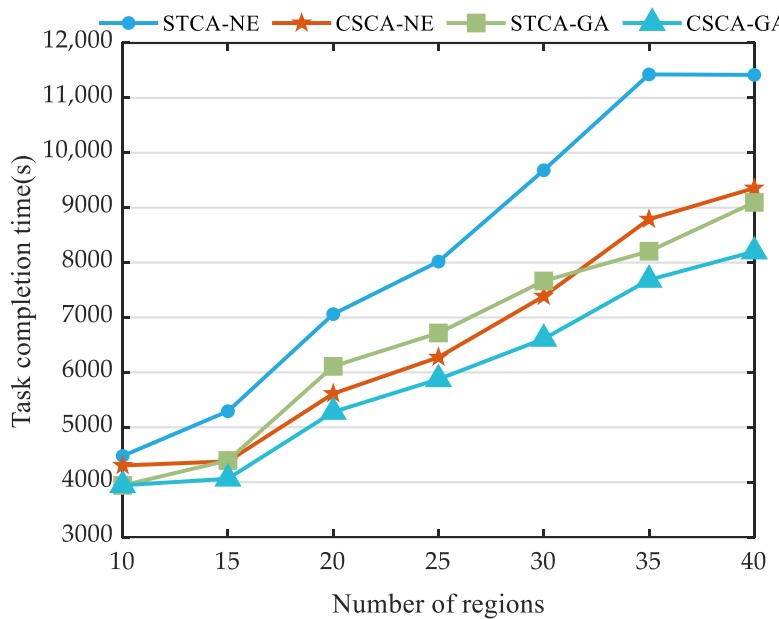

**Figure 6.** Comparison of task completion time achieved using the four algorithms when the region number increases from 10 to 40.

**Table 3.** The statistical results (in seconds) of task completion time under different region numbers.

| Numbers | STCA-NE | CSCA-NE | STCA-GA | CSCA-GA |
|---------|---------|---------|---------|---------|
| 10 | 4479 | 4304 | 3942 | 3942 |
| 15 | 5293 | 4380 | 4395 | 4063 |
| 20 | 7061 | 5613 | 6111 | 5279 |
| 25 | 8018 | 6275 | 6712 | 5875 |
| 30 | 9682 | 7387 | 7665 | 6603 |
| 35 | 11,428 | 8789 | 8206 | 7706 |
| 40 | 11,419 | 9357 | 9096 | 8201 |

In Figure 7, the number of regions is set to 20, and we attempt to compare the results of the four algorithms under different numbers of UAVs. To better understand the results, the statistical results of task completion time obtained using different algorithms are shown in Table 4. It can be seen that with the increase of UAVs, the overall task completion time of the four methods shows a downward trend. In addition, under different numbers of UAVs, the average task completion time calculated using CSCA-NE is 21.1% lower than STCA-NE, and CSCA-GA is 13.1% lower than STCA-GA. The results of CSCA-NE even outperform those of STCA-GA consistently throughout the entire process, with the exception of UAVs; the number is four. Furthermore, it is apparent that there is a minimal disparity in outcomes between CSCA-NE and CSCA-GA, with their average difference being 60.4% lower than the average difference between STCA-NE and STCA-GA. This reinforces the notion of the CSCA algorithm's remarkable adaptability in sequentially allocating regions in clustering.

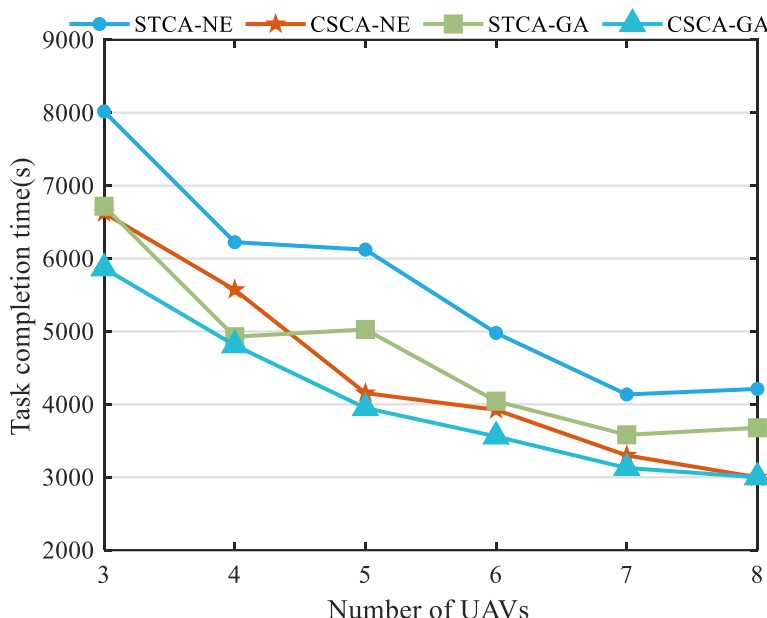

**Figure 7.** Comparison of task completion time achieved using the four algorithms when the UAVs number increases from 3 to 8.

**Table 4.** The statistical results (in seconds) of task completion time under different UAV numbers.

| Numbers | STCA-NE | CSCA-NE | STCA-GA | CSCA-GA |
|---------|---------|---------|---------|---------|
| 3 | 8018 | 6625 | 6718 | 5866 |
| 4 | 6224 | 5572 | 4928 | 4811 |
| 5 | 6122 | 4155 | 5028 | 3950 |
| 6 | 4979 | 3924 | 4044 | 3560 |
| 7 | 4136 | 3300 | 3582 | 3128 |
| 8 | 4211 | 2999 | 3678 | 2999 |

*5.3. Execution Time Simulation*

This section further compares the differences in calculation time between algorithms based on the previous subsection. Figure 8 demonstrates the comparative results of simulation time for four algorithms as the number of regions increases from 10 to 40. Table 5 presents the statistical results of the time consumed by various algorithms for different numbers of regions. The computation time for CSCA-GA and STCA-GA is within the same order of magnitude, around 0.1 s, while the computation time for CSCA-NE and STCA-NE is in the millisecond range. In Table 5, despite STCA-NE having the shortest computation time, the maximum difference between STCA-NE and CSCA-NE is approximately 3 milliseconds. The computation time of CSCA-NE and STCA-NE is significantly lower than that of CSCA-GA and STCA-GA due to the higher complexity of GA in contrast to the nearest-end strategy. Moreover, the increase in the number of regions displays a gradual rise in computation time, suggesting a minimal impact of the number of regions on the algorithm's computation time.

Table 6 displays the time consumed by various algorithms as the number of UAVs increases from 3 to 8; Figure 9 is a graphical representation of Table 6. As can be seen in Figure 9, the execution times of CSCA-GA and STCA-GA remain highly similar, and this similarity also holds for CSCA-NE and STCA-NE. The calculation times for both CSCA-NE and STCA-NE scarcely increase with an increasing number of UAVs; however, CSCA-GA and STCA-GA exhibit a gradual increase. This occurs because the increase in UAVs results in a rise in the number of calls made to GA, which, in turn, results in an increase in time. The computation time of STCA-NE is the shortest, while the computation time of CSCA-NE

is also within the same scale of magnitude as STCA-NE, with a maximum difference of 5 milliseconds between them.

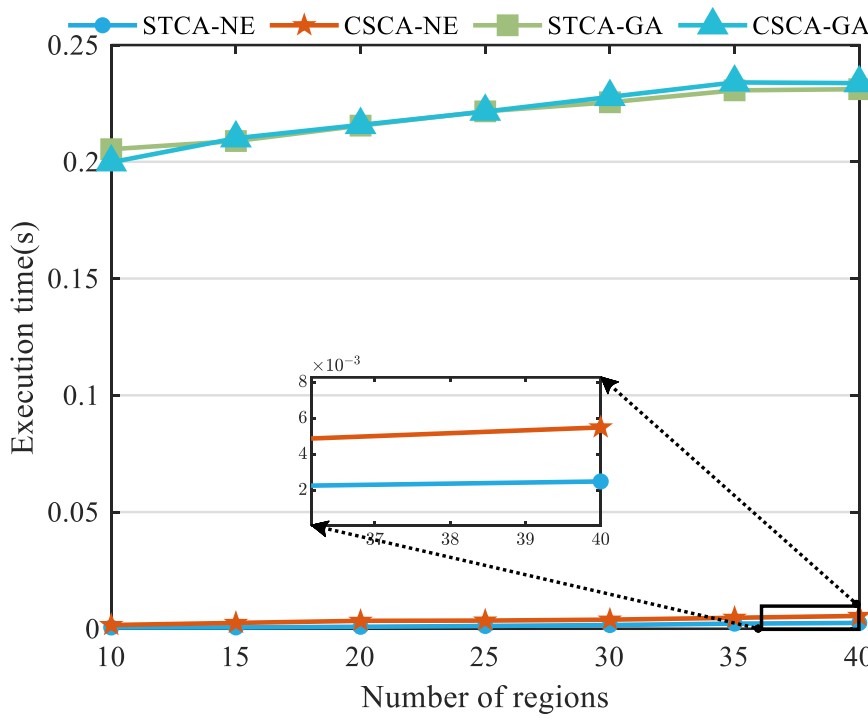

**Figure 8.** Comparison of execution time among the four algorithms when the region number increases from 10 to 40.

**Table 5.** The statistical results (in seconds) of the time consumed using various algorithms under different region numbers.

| Numbers | STCA-NE | CSCA-NE | STCA-GA | CSCA-GA |
|---------|---------|---------|---------|---------|
| 10 | 0.00055 | 0.00160 | 0.20540 | 0.19980 |
| 15 | 0.00058 | 0.00250 | 0.20890 | 0.21010 |
| 20 | 0.00085 | 0.00340 | 0.21550 | 0.21580 |
| 25 | 0.00120 | 0.00350 | 0.22160 | 0.22150 |
| 30 | 0.00160 | 0.00390 | 0.22550 | 0.22780 |
| 35 | 0.00220 | 0.00470 | 0.23060 | 0.23400 |
| 40 | 0.00250 | 0.00550 | 0.23110 | 0.23370 |

**Table 6.** The statistical results (in seconds) of the time consumed by various algorithms under different UAV numbers.

| Numbers | STCA-NE | CSCA-NE | STCA-GA | CSCA-GA |
|---------|---------|---------|---------|---------|
| 3 | 0.00140 | 0.00370 | 0.22350 | 0.22180 |
| 4 | 0.00110 | 0.00400 | 0.28540 | 0.29150 |
| 5 | 0.00110 | 0.00440 | 0.35270 | 0.34950 |
| 6 | 0.00100 | 0.00510 | 0.40410 | 0.41290 |
| 7 | 0.00091 | 0.00560 | 0.46260 | 0.46020 |
| 8 | 0.00085 | 0.00600 | 0.51740 | 0.52510 |

All of the above results demonstrate that, when compared to the STCA algorithm, the CSCA algorithm proposed in this work can achieve more accurate results while ensuring the computational speed at the same scale, and it has better adaptability for optimizing subsequent access sequences.

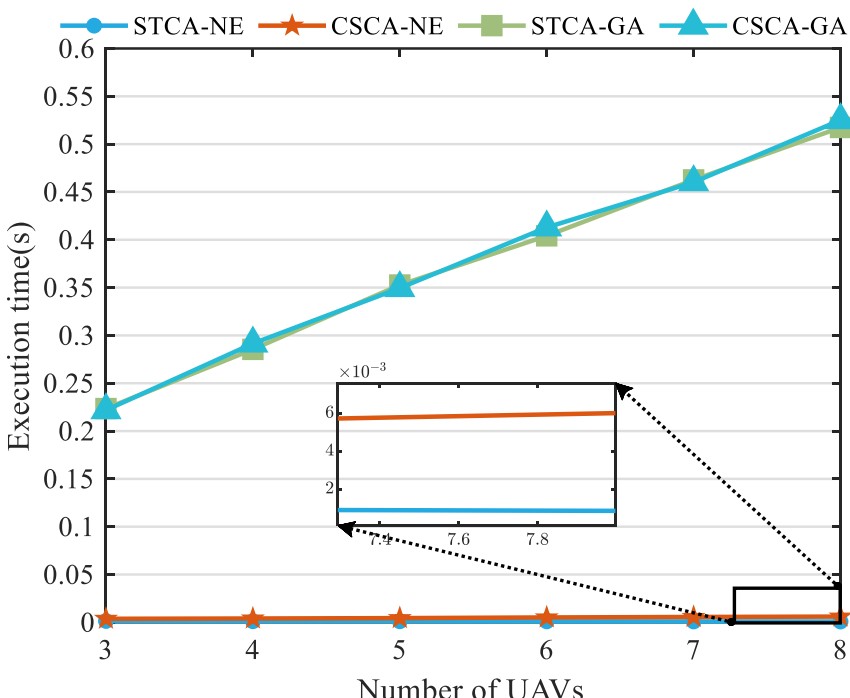

**Figure 9.** Comparison of execution time among the four algorithms when the UAV number increases from 3 to 8.

### 5.4. Flighting-Path Simulation

In this section, the top three UAVs from Table 2 will be selected, and the number of regions will be set to 15. In order to enhance the visibility of simulation results, the range of values for the long axes of the regions is [6–8], while the range for the short axes is [4, 4.5, 5]. Furthermore, the minimum turning radius constraint of fixed-wing UAVs is considered. In this simulation, the minimum turning radius of each UAV is set to 200 m, and the flight trajectory is optimized using the Dubins curve optimization method [37]. Figure 10 shows the allocation sequence obtained using the CSCA-GA algorithm, and it displays UAVs represented by lines of different colors, with black arrows indicating their respective flight directions.

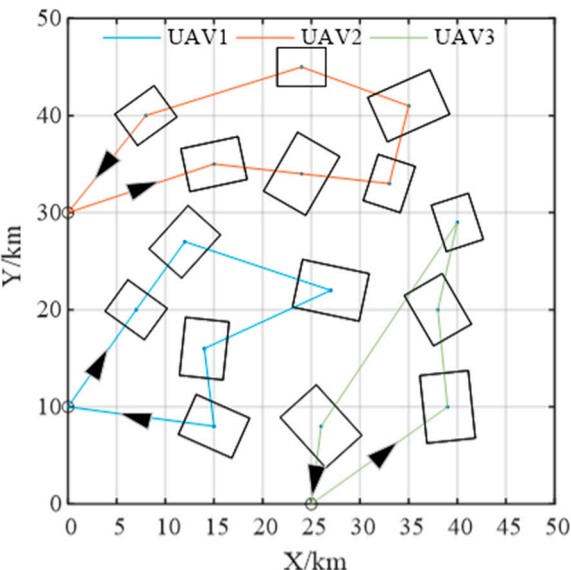

**Figure 10.** Diagram of UAVs executing tasks in sequential order.

The flight trajectories of the UAVs obtained using the BSSS algorithm and long edge scanning strategy (in this study called LESS) are depicted in Figures 11a and 11b, respectively. Upon comparing the two figures, it becomes evident that the UAV employs distinct scanning methods for the area enclosed by the red dashed circle in each figure. Unlike LESA, BSSS does not carry out scanning along the longer side. In contrast to scanning along the longer side, BSSS achieves a reduction of approximately 8 km in total distance compared to LESA. As a result, this also substantiates the superiority of the proposed BSSS, as presented in this study.

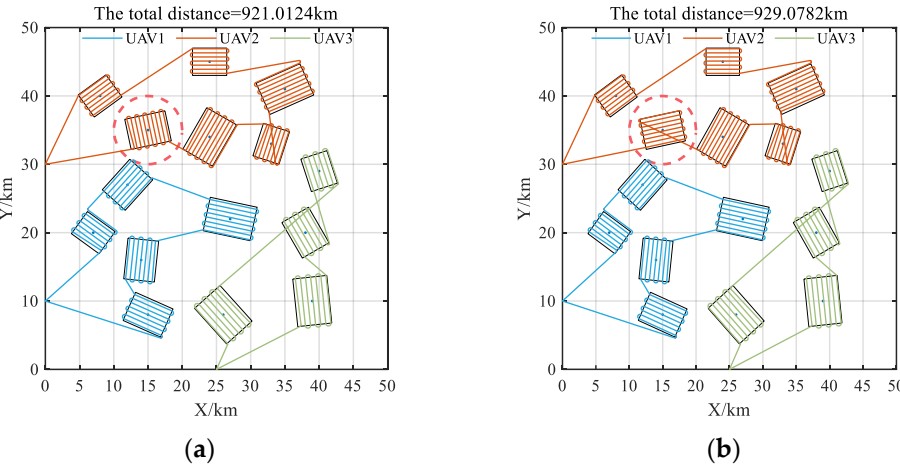

**Figure 11.** Illustrative diagram of UAVs flight path: (**a**) UAVs flight path based on BSSS; (**b**) UAVs flight path based on LESS.

## 6. Conclusions

This study researched the multi-region coverage path-planning problem to identify appropriate region coverage schemes for UAVs with varying flight speeds and scanning widths. The main conclusions are as follows:

1. A novel clustering algorithm was proposed to assign regions to UAVs, which effectively reduces the computational complexity during the task allocation process;
2. A regional iterative strategy was designed to ensure the balancing of task completion time among UAVs;
3. A path-planning method was devised to select the entry points and scanning modes of the regions by considering the shortest flight distance, offering a reference trajectory for the actual flight of UAVs;
4. The simulation results demonstrated that the clustering algorithm (CSCA) proposed in this study surpasses others in terms of both quality and flexibility. Compared to utilizing STCA for clustering, the average task completion time of CSCA has decreased by 9–21%. In addition, the discrepancy between CSCA-NE and CSCA-GA was approximately 60% lower than the discrepancy between STCA-NE and STCA-GA. Furthermore, the path-planning method (BSSS) proposed in this study can yield shorter flight paths.

Although the method proposed in this study has the capability of swiftly achieving results within a brief timeframe, we also want to enhance the algorithm's accuracy by increasing the coupling between subproblems, which will be the primary focus of our future research.

**Author Contributions:** Conceptualization, P.X. and N.L.; methodology, P.X. and N.L.; software, P.X.; validation, P.X. and N.L.; formal analysis, P.X.; investigation, P.X.; resources, P.X. and N.L.; data curation, P.X.; writing—original draft preparation, P.X.; writing—review and editing, P.X. and N.L.; visualization, P.X.; supervision, N.L., F.X., H.N., M.Z. and B.W.; project administration, N.L.; funding acquisition, N.L. All authors have read and agreed to the published version of the manuscript.

**Funding:** This research was funded by the National Natural Science Foundation of China (No. 62003272, 52372398 & 62003266).

**Data Availability Statement:** Not applicable.

**Conflicts of Interest:** The authors declare no conflict of interest.

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
