# Peer review of "Clustering-Based Multi-Region Coverage-Path Planning of Heterogeneous UAVs"

_drones, doi:10.3390/drones7110664_

Round 1
Reviewer 1 Report
Comments and Suggestions for Authors
In this paper, a clustering-based approach to the multi-region coverage path planning issue of heterogeneous UAVs was investigated. Prior to developing a planning formula based on an integer linear programming model, it was required to assess the restrictions that would be necessary during the planning process. This problem was subsequently broken down into the sub-problems of optimizing visiting orders and geographic allocation. The authors put forth a brand-new clustering approach that uses the nearest-to-end policy to optimize the visiting order along with centroid iteration and spatiotemporal similarity to allocate regions. They also suggested a method for creating region scanning trajectories, which serve as trajectory models for actual flight, called distance-based bilateral shortest selection. The effectiveness of the clustering algorithm and region-scanning approach suggested in this paper is demonstrated by simulation results. As a result, the work is worthy of publication if the remarks listed are taken into account:
1- There are numerous similar types of research that address the UAVs’ path planning like robot ecosphere-based ones. Hence please explain why clustering-based multi-region coverage is chosen deeply.
2- Additionally, there are similar works published before like “A Clustering-Based Coverage Path Planning Method for Autonomous Heterogeneous UAVs”, and “An Adaptive Clustering-Based Algorithm for Automatic Path Planning of Heterogeneous UAVs”. Hence, what is new is not clear to me. An additional explanation is necessary.
3- Even if the authors have investigated the previous studies I think it is still not clear the gap between the previous studies and the current one. Hence, the authors should clarify the gap between the existing research work and the work you intend to do.
4- The structure of this article can be strengthened. Please draw a flow chart of this article and place it at the end of the Section 1 Introduction.
5- The authors have to clearly state the limitations of their study.
6- There are different types of path planning algorithms like "the Generalized Voronoi Diagrams (GVD)", "a Rapidly Exploring Random Tree (RRT)", and "the Gradient Descent Algorithm (GDA)". Hence please use at least one of the aforementioned algorithms and compare the results. This will improve the quality of the paper.
7- It is common knowledge that Dijkstra's method discovers the shortest path between each node (up until the target is reached) starting at a root node. One of the most helpful graph algorithms is Dijkstra, which is also easily adaptable to address a wide range of issues. Therefore, kindly explain the rationale behind the selection of your approach and, if at all feasible, repeat the analyses using Dijkstra's method.
8- The following articles can be considered and added to the introduction part of the study to improve the quality of the study. 1) A clustering-based coverage path planning method for autonomous heterogeneous UAVs; 2) BOLD Bio-Inspired Optimized Leader Election for Multiple Drones; 3) Analysis of Wavelet Controller for Robustness in Electronic Differential of Electric Vehicles An Investigation and Numerical Developments; 4) Configurations and applications of multi-agent hybrid drone/unmanned ground vehicle for underground environments: a review; 5) Design optimization of a fixed-wing aircraft; 6) Intelligent Autonomous Systems 13; 7) Stereo Visual Inertial Mapping Algorithm for Autonomous Mobile Robot; 8) Computer Vision – ECCV.
9- The authors' use of "we..." in sentence construction is unprofessional and appears throughout the manuscript. Alternately, substitute "it...." for the alleged structure.
10- The conclusion should be written in bullet points with only the most important findings.
11- Finally, while the work is well-written in general, it, unfortunately, contains some grammatical and typographical problems. Before resubmitting the manuscript, it is suggested that the authors reread it.
Comments on the Quality of English Language
While the work is well-written in general, it, unfortunately, contains some grammatical and typographical problems. Before resubmitting the manuscript, it is suggested that the authors reread.
Reviewer 2 Report
Comments and Suggestions for Authors
This study dives into the challenges of planning flight paths for different types of drones, known as heterogeneous UAVs, when they have to cover multiple areas. Since these drones have varied capabilities, plotting their routes isn't straightforward and can get quite complex. To simplify this, the researchers came up with a method that breaks down the problem: first, deciding which drone goes where, and then, in what order they should visit these places. They used a mathematical approach to make this planning more efficient. The end goal? To have these drones cover their assigned areas in the shortest time possible.
Comments:
1. Introduction:
· The intro talks about UAVs in general, but I'd love to hear more about the unique challenges when dealing with different types of UAVs.
· Also, how does this study stand out from others? A bit more clarity would be great.
2. System Model:
· The authors described the different UAV types well, but how do these differences impact the missions? Maybe some real-life examples would help.
3. Problem Description:
· Why only fixed-wing UAVs? Are they better for this kind of work?
· The authors mention "heterogeneity" a lot, but diving deeper into what that means in practice would be super helpful.
4. Exact Formulation:
· The constraints they've listed are clear, but how do they play out in the real world? Are there challenges we should know about?
· The authors want to minimize completion time, but what about other factors like energy use?
5. Proposed Methods:
· Their new clustering method sounds cool, but are there situations where it might not work as well?
· A simple breakdown or even a diagram of their strategy would make things clearer.
6. Simulation Results:
· It's great they tested their methods, but how do they stack up against other approaches?
· What did they specifically measure in their tests? Are there real-world factors that might change the results?
7. Conclusion:
· The authors say their method is better than others, but some numbers or comparisons would back that up nicely.
· What's next? Any ideas for future research or real-world uses?
8. To wrap up, the study has some solid points about planning paths for different UAVs. But, a bit more detail in some areas and some comparisons to other methods would round it out.
9. There are some English notes that can be identified in the study:
· Some sentences sound a bit off or too formal. Like when they say "UAVs' heterogeneity mainly lies in..." – it'd be clearer as "The main differences between the UAVs are their flight speed, endurance, and scanning width."
· Some sentences are super long. Breaking them up might make things clearer.
· They mention "The constraint on the scanning of regions." It'd be clearer as "To avoid doing things twice, each task region should only be scanned by one UAV."
· There's a bit that says "The constraint on the number of UAV engaged in all tasks." It should be "UAVs" (plural) instead of "UAV."
· When they say "The time consumption of U i in scanning R j", it'd sound better as "How long U i takes to scan R j."
· They've got "The set of the clusters" – dropping "the" to make it "The set of clusters" sounds smoother.
· And there's "The characteristics of each region in this study are denoted as..." which could be "In this study, each region's features are shown as..."
10. The authors should make table for evaluation parameters used in the simulation.
11. What are the limitations of proposed method? Please answer the question in the article.
12. Please follow the journal template.
Comments on the Quality of English Language
9. There are some English notes that can be identified in the study:
· Some sentences sound a bit off or too formal. Like when they say "UAVs' heterogeneity mainly lies in..." – it'd be clearer as "The main differences between the UAVs are their flight speed, endurance, and scanning width."
· Some sentences are super long. Breaking them up might make things clearer.
· They mention "The constraint on the scanning of regions." It'd be clearer as "To avoid doing things twice, each task region should only be scanned by one UAV."
· There's a bit that says "The constraint on the number of UAV engaged in all tasks." It should be "UAVs" (plural) instead of "UAV."
· When they say "The time consumption of U i in scanning R j", it'd sound better as "How long U i takes to scan R j."
· They've got "The set of the clusters" – dropping "the" to make it "The set of clusters" sounds smoother.
· And there's "The characteristics of each region in this study are denoted as..." which could be "In this study, each region's features are shown as..."
Reviewer 3 Report
Comments and Suggestions for Authors
The authors presented a UAV multi-area coverage path planning. The paper is well-written and well-divided. However, I have a few comments and queries:
-The CSCA-NE task completion time exceeds that of STAG-GA after 35 regions. Can the authors comment on this?
-The authors certainly need to explain a little better how they arrived at equations (1) and (2).
-The authors could comment further in the conclusion. They could provide a brief quantitative summary to demonstrate the benefits of the proposed technique.
-The work has been written in great detail, and the contribution is very clear. I have no further comments.
Round 2
Reviewer 1 Report
Comments and Suggestions for Authors
The authors have taken into account nearly every suggestion, with the exception of the proposed articles, which, in my opinion, are all relevant and enhance the study's background. As such, I ask that you reconsider the suggested papers and make any required adjustments. Additionally, there are still grammatical/typos mistakes, hence please reread the paper and correct grammatical/typos errors.
Reviewer 2 Report
Comments and Suggestions for Authors
All the corrections and suggestions are made correctly.
Good Luck
Author Response
We greatly appreciate your valuable comments on this paper!